# Longitudinal Study of the Association between General Anesthesia and Increased Risk of Developing Dementia

**DOI:** 10.3390/jpm11111215

**Published:** 2021-11-16

**Authors:** Jong-Hee Sohn, Jae Jun Lee, Sang-Hwa Lee, Chulho Kim, Hyunjae Yu, Young-Suk Kwon, Dong-Kyu Kim

**Affiliations:** 1Department of Neurology, Chuncheon Sacred Heart Hospital, Hallym University College of Medicine, Chuncheon 24253, Korea; deepfoci@hallym.or.kr (J.-H.S.); neurolsh@hallym.or.kr (S.-H.L.); gumdol52@hallym.or.kr (C.K.); 2Division of Big Data and Artificial Intelligence, Institute of New Frontier Research, Chuncheon Sacred Heart Hospital, Hallym University College of Medicine, Chuncheon 24253, Korea; iloveu59@hallym.or.kr (J.J.L.); yunow@hallym.or.kr (H.Y.); 3Department of Anesthesiology and Pain Medicine, College of Medicine, Chuncheon Sacred Heart Hospital, Hallym University College of Medicine, Chuncheon 24253, Korea; 4Department of Otorhinolaryngology-Head and Neck Surgery, Chuncheon Sacred Heart Hospital, Hallym University College of Medicine, Chuncheon 24253, Korea

**Keywords:** general anesthesia, dementia, Alzheimer’s disease, vascular dementia

## Abstract

The association between exposure to general anesthesia (GA) and the risk of dementia is still undetermined. To investigate a possible link to the development of dementia in older people who have undergone GA, we analyzed nationwide representative cohort sample data from the Korean National Health Insurance Service. The study cohort comprised patients over 55 years of age who had undergone GA between January 2003 and December 2004 and consisted of 3100 patients who had undergone GA and 12,400 comparison subjects who had not received anesthesia. After the nine-year follow-up period, we found the overall incidence of dementia was higher in the patients who had undergone GA than in the comparison group (10.5 vs. 8.8 per 1000 person-years), with the risk being greater for women (adjusted HR of 1.44; 95% CI, 1.19–1.75) and those with comorbidities (adjusted HR of 1.39; 95% CI, 1.18–1.64). Patients who underwent GA showed higher risks for Alzheimer’s disease and vascular dementia (adjusted HR of 1.52; 95% CI, 1.27–1.82 and 1.64; 95% CI, 1.15–2.33, respectively). This longitudinal study using a sample cohort based on a nationwide population sample demonstrated a significant positive association between GA and dementia, including Alzheimer’s disease and vascular dementia.

## 1. Introduction

As the population ages, surgery is being performed more frequently and in progressively older adults [1]. Cognitive dysfunction is relatively common during the postoperative course of older adults, and anesthetics have been thought to be a possible contributor [2]. Postoperative cognitive dysfunction (POCD) after anesthesia and surgery is a common severe complication, especially in older patients. The one-week and three-month incidence of POCD in surgical patients aged over 60 years are 25.8–41.4% and 9.9–12.7%, respectively [3,4]. There is a perception that POCD may increase the risk of dementia and Alzheimer’s disease (AD). Dementia is the most common age-related disorder of the brain characterized by progressive cognitive and functional declines. AD is the most prevalent type of dementia. Its predominance is expected to increase as the aged population expands. Dementia imposes significant economic burdens on society. A better understanding of the relationship between dementia and surgery with anesthesia is critical.

In vivo and in vitro studies suggest that anesthetics increase brain AD pathology, although such effects seem to be mediated by only certain anesthetics. Animal models support this perception by demonstrating associations between GA exposure and AD pathogenesis. Inhaled or intravenous anesthetics could promote the formation of Aβ plaques and neurofibrillary tangles [5,6,7,8,9,10]. Cell models have found that inhaled anesthetics induced apoptosis and increased β-amyloid protein levels. Commonly used inhaled anesthetics may promote AD neuropathogenesis [11,12]. Multiple anesthetic agents trigger hyperphosphorylation of the microtubule-associated protein tau, which forms neurofibrillary tangles in AD patients [7,13]. However, the impact of GA on cognitive impairment including dementia is controversial and complex. Despite previous experimental reports, findings from human studies have not been consistent. In some previous studies, exposure to GA was observed to be significantly associated with an increased risk of dementia or AD [14,15,16,17,18], but in others no association was found [19,20,21,22,23,24,25,26]. Nationwide population-based cohort studies found significant associations between GA and dementia [17,27]. However, previous meta-analyses yielded conflicting results [28,29,30]. One meta-analysis reported a significant positive association between GA and AD [30], but others found only weak or no links [28,29]. Thus, any association between anesthesia and dementia remains unclear. Further investigation is needed.

Therefore, we evaluated the risk of dementia a large population-based cohort that had undergone GA. We used a large dataset provided by the National Health Insurance Service (KNHIS) of South Korea. We analyzed the risk of dementia, including AD and vascular dementia (VD), in patients who have undergone GA to identify GA as a potential risk factor using nationwide representative cohort sample data in the present study.

## 2. Materials and Methods

### 2.1. Korea National Health Insurance Service

This study adhered to the tenets of the Declaration of Helsinki and used data from the national health claims database collected by the KNHIS. It was approved by the Institutional Review Board of Hallym Medical University, Chuncheon Sacred Hospital (IRB No. 2021-08-006), and the need for written informed consent was waived as the KNHIS dataset used in the study comprised de-identified secondary data. KNHIS employs the Korean Classification of Diseases (KCD), which is very similar to the International Classification of Diseases, as a system of diagnostic practice codes. Here, we used a national sample cohort (NSC) dataset collected from 2002 to 2013 containing information for 1,025,340 representative random subjects and accounting for approximately 2.2% of the South Korean population in 2002 (46 million). Stratified random sampling was performed using 1476 strata with respect to age (18 groups), sex (two groups), and income level (41 groups: 40 health insurance and one medical aid beneficiary).

### 2.2. Study Population

We conducted a retrospective cohort study using the KNHIS-NSC database. The study cohort consisted of patients who underwent GA during the index period (1 January 2003, to 31 December 2004), and were over 55 years of age at cohort entry. To remove any potential pre-existing cases of GA, we established a washout period (1 January 2002, to 31 December 2002). Additionally, we excluded patients who: (1) underwent GA before or after the index period; (2) underwent other types of anesthesia from 2002 to 2013; (3) had a history of brain or heart surgery from 2002 to 2013; (4) were diagnosed with dementia before the index period; or (5) died of any cause from 2002 to 2013. The comparison group (patients who did not undergo anesthesia) comprised randomly selected propensity score–matched individuals from the remaining cohort registered in the database (four for each patient who underwent GA) between 2003 and 2004. Variables used in the process of the selection of the matched sample included sociodemographic factors (age, sex, residential area, and household income), comorbidities, and the enrolled date. The schematic description of the cohort design is presented in Figure 1. All enrolled patients were tracked until 2013, and patients diagnosed with dementia (AD (F00, G30), VD (F01), others (F02, F03)) were identified. All patients who experienced no events and were alive until 31 December 2013, were censored after this time point (Table A1).

### 2.3. Predictor and Outcome Variables

The study population was divided into three age groups (55–64, 65–74, ≥75 years), three income groups (low: ≤30%, middle: 30.1–69.9%, and high: ≥70% of the median), and three residential areas (1st area: Seoul, the largest metropolitan region in South Korea; 2nd area: other metropolitan cities in South Korea; and 3rd area: small cities and rural areas). The operational definitions of study endpoints were all-cause mortality or the incidence of dementia. All patients who had no events and were alive until 31 December 2013 were censored after this time point. The risks of dementia were compared between patients who underwent GA and comparison (without anesthesia) groups by using person-years at risk, which were defined as the duration between either the date of GA or the date of enrolment (for the comparison group) and the patient’s respective endpoint. In this study, comorbidities, including hypertension (I10–15), diabetes mellitus (E10–14), stroke (I60–63), and disorders of lipoprotein metabolism and other lipidemias (E78), were obtained based on the diagnostic code of the KNHIS dataset.

### 2.4. Sensitivity Test

We selected patients who were diagnosed with dementia (AD (F00, G30), VD (F01), and others (F02, F03)) over 65 years of age from this dataset during 2012–2013. For the matched comparison group, we selected patients who had not been diagnosed with dementia (through 2002–2013) at age over 65 years from this dataset during 2012–2013. The comparison group (two patients for every one patient with dementia) was obtained by using propensity score matching, according to age, sex, residential area, household income, and comorbidities (Table A2). Finally, using the analysis of sensitivity test results, we excluded all patients who had a history of brain or heart surgery between 2002 and 2013.

### 2.5. Statistical Analysis

We employed propensity score-matching according to age, sex, residential area, household income, and comorbidities. Incidence rates per 1000 person-years for dementia were obtained by dividing the number of patients with dementia by person-years at risk. The overall disease-free survival rate was determined using Kaplan–Meier survival curves for the entire observation period. To identify whether GA increased the risk of occurrence of dementia, we used Cox proportional hazard regression to calculate the hazard ratio (HR) and 95% confidence interval (CI), adjusting for other predictor variables. The Levene test was performed to test homogeneity, and the Welch analysis of variance was completed to investigate the difference in the frequency of GA. All statistical analyses were performed using R version 3.3.1 (R Foundation for Statistical Computing, Vienna, Austria) with a significance level of 0.05.

## 3. Results

### 3.1. Effect of General Anesthesia on the Incidence of Dementia in Patients Older Than 55 Years

The present study consisted of 3100 patients who underwent GA and 12,400 comparison subjects (patients who did not undergo anesthesia). The characteristics of the study population for the two cohorts (the GA group and the comparison group) are presented in Table 1. There were similar distributions of sex, age, residential area, household income, and comorbidities between the two groups because each variable was matched appropriately (Figure 2). We found that the overall incidence of dementia was higher in the patients who underwent GA (10.5 per 1000 person-years) than in the comparison group (8.8 per 1000 person-years) (Table 2).

### 3.2. Hazard Ratios for Dementia in Patients Aged over 55 Years Who Underwent General Anesthesia

To analyze the HR for the development of dementia, we used simple and multiple Cox regression models (Table 2). After adjusting for sociodemographic factors and comorbidities, the GA group was found to be associated with prospective dementia development with an adjusted HR of 1.36 (95% CI, 1.16–1.58). In addition, we found that increasing age, lower household income, and comorbidities were significantly associated with the prospective development of dementia. Figure 3 presents the Kaplan–Meier survival curves with log-rank tests for the cumulative hazard plot of dementia between the comparison and GA groups. The results of the log-rank test indicated that patients who underwent GA developed dementia more frequently than patients who did not undergo anesthesia during the nine-year follow-up period.

On subgroup analysis, we observed a significant likelihood of dementia development in females (adjusted HR of 1.44; 95% CI, 1.19–1.75) and subjects with comorbidities (adjusted HR of 1.39; 95% CI, 1.18–1.64) (Table 3 and Table 4). Moreover, we detected that the adjusted HR of developing AD and VD in patients ≥55 who underwent GA during the nine-year follow-up period was 1.52 (95% CI, 1.27–1.82) and 1.64 (95% CI, 1.15–2.33), respectively, compared with the comparison group (Table 5). The Kaplan–Meier survival curves with log-rank tests for AD and VD during the nine-year follow-up period are presented in Figure 4.

### 3.3. Sensitivity Test for Effect of General Anesthesia on the Incidence of Dementia

To determine whether our findings reflected an incidental temporal relation between GA and dementia, we performed sensitivity analyses. From 2012 to 2013, we retrospectively counted the frequency of GA events between the dementia and non-dementia groups to 2002 (Table 6). The dementia group experienced significantly more frequent events of GA compared to the comparison group (*p* < 0.001) (Table 7). Thus, this test could support our findings regarding the association of GA with increased dementia.

## 4. Discussion

The present study investigated the incidence of dementia in middle-aged and older patients (≥55 years) who had undergone GA and compared them with matched subjects who had not undergone anesthesia using KNHIS-NSC data. The overall incidence of dementia during the nine-year follow-up period was higher in the patients who underwent GA than in the comparison group (10.5 vs. 8.8 per 1000 person-years). We found that the GA group was associated with prospective dementia development, with the risk being greater for female patients and those with combined comorbidities. Moreover, the patients who underwent GA showed higher risks for both AD and VD.

POCD is a common severe complication after GA, especially in older patients. POCD is characterized by a loss of attention, memory impairment, and personality changes that persist for months or years. However, a link to other cognitive disorders such as dementia has not been made clear. Despite some experimental data obtained in cell and animal models suggesting a relationship between anesthesia and neurotoxicity in older adults, a definitive link remains elusive in humans [9]. Some experimental studies suggest that anesthetic exposure impairs cognitive function and increases pathology commonly associated with AD. Some in vitro [5,11,31] and animal [11,13,32] studies have demonstrated increased production and aggregation of β-amyloid peptides. Although these experimental data suggested a significant association between GA and AD, evidence from human clinical studies is controversial.

Previous observational studies found either an increased risk of developing dementia following exposure to GA [17,18,33] or no increase in risk [24,26,34]. Two previous meta-analyses [17,29] and a reanalysis [35] reported no evidence of a significant relationship between GA and dementia. Another meta-analysis, which included more recently published literature, demonstrated a significant positive association between GA and AD [30]. Our longitudinal study using a sample cohort based on a nationwide population suggested that GA can increase the risk of dementia over a long period. By contrast, large, retrospective, population-based, nested case-control and prospective cohort studies found a lack of association between GA and dementia [26,34]. However, these studies were limited by small samples and short follow-up duration. Similar to our study, one nationwide retrospective cohort study in Taiwan reported that patients undergoing GA showed an increased risk of developing dementia [18]. Additionally, another nationwide population-based cohort study in South Korea reported a significant association between GA and dementia [27].

In our study, the GA group was found to be associated with prospective dementia development with an adjusted HR of 1.36 after adjusting for sociodemographic factors and comorbidities. In subgroup analysis, a significant increase of the risk of dementia was found in females and in subjects with comorbidities. Another large retrospective cohort study comparing outcomes in patients undergoing GA as compared with nonsurgical controls found adjusted HRs of 1.29 and 1.46 of developing dementia in those who underwent GA [18,27], similar to the results of our study. In subgroup analysis of these studies, one study found increased risk of dementia in patients who had received multiple anesthesia agents and who had had multiple exposures to GA [36], whereas another study found no significant difference in the incidence of dementia between patients with exposure to anesthesia at least twice and exposure only once within a year [18]. However, our study had a unique advantage compared with other nationwide cohort studies. In this study, we performed the sensitivity test to confirm whether our findings reflected an incidental temporal relation between GA and dementia. The result of the sensitivity test supported our findings regarding the association between GA and dementia. Further research is needed on the relationship between the number of exposures to anesthesia and the risk of dementia.

Moreover, we detected a high risk of developing VD in addition to AD in patients ≥ 55 years of age who underwent GA during the nine-year follow-up period, compared with the comparison group. A previous study examined differences in risk factors and with regard to behavior in subjects with VD and probable AD. VD subjects were more likely to have a history of GA, and the authors concluded that GA is a risk factor for VD [37]. Recent clinical studies have shown that preoperative white matter lesions, produced by chronic cerebral hypoperfusion, are frequently observed in older people and are a serious and significant risk factor for postoperative delirium and POCD [38,39,40]. Previous experimental studies also demonstrated that persistent hypocapnea or hypotension caused neuronal damage in the caudoputamen or the hippocampus in a rat model of chronic cerebral hypoperfusion, which features global cerebral white matter lesions without neuronal damage and is recognized as a good model of human VD, especially in older people [41,42]. Thus, it was suggested that, in addition to the possible proposed mechanisms of neuroinflammation and neurodegeneration, i.e., amyloid β accumulation and/or tau protein phosphorylation, perioperative vital sign changes that cause reductions in cerebral blood flow might contribute to POCD in patients with white matter lesions, whose cerebral blood flow is already considerably decreased [43,44]. However, studies on the relationship between VD and GA are lacking, and further studies are needed to determine the relationship between these two factors in the future.

Moreover, typical sporadic AD is likely to be driven by a complex interplay between genetic and environmental factors that develops over decades. It is now thought that ~70% of AD risk is attributable to genetic factors [45]. The pathophysiological process of AD is thought to begin many years before the diagnosis of AD dementia. Emerging evidence from both genetic at-risk and aging cohorts suggests that there is a time lag of a decade or more between the beginning of the pathological cascade of AD and the onset of clinically evident impairment [46]. A low pre-operative cognitive reserve may affect the link between anesthesia and long-term or persistent POCD [47,48]. Preclinical in vivo studies found that pre-symptomatic AD transgenic mice evidenced more cognitive dysfunction after surgery than did controls [49]. Clinical studies also found that pre-existing brain dysfunction was a risk factor for post-operative delirium and POCD, despite the absence of any obvious residual neurological deficits at the time of surgery [4,50]. Therefore, although our analysis excluded individuals with dementia diagnosed before and during the index period or with a long follow-up period, we do not have detailed information on the cognitive function or brain imaging of individuals before GA, so it is possible that there may have been baseline differences between the GA exposure and comparison groups in cognition or preclinical AD or VD pathology that are not captured in our databases. Moreover, a baseline difference probably exists between patients who underwent GA and not. The former exhibited pathologies that necessitated GA. We found it difficult to evaluate the effects of various surgeries, intra-operative events, and perioperative complications, such as hypoxia and hemodynamic instability. Although no effect of surgery per se on dementia risk was established, high-risk surgeries (such as cardiac surgery) have been reported to increase the risk of cognitive impairments, including delirium and postoperative cognitive dysfunction [51,52]. To overcome these limitations, it is necessary to evaluate the outcomes in patients with the same pathologies who underwent local anesthesia and GA.

Additionally, our study has several limitations. This study had a retrospective design, and thus there is the potential that unmeasured confounders may not have been accounted for. This study design would not be able to separate the effects of GA exposure from the effects of surgical stress and other potential confounders regarding pre- or post-operative situations. The KNHIS–NSC dataset focuses on medical claims and reimbursements. This is not a research dataset. The representative, nationwide, population-based dataset contains information on the medical service utilization of more than 1 million Koreans. A total of 1,025,340 participants of the cohort, 2.2% of the total eligible population, were randomly sampled from the 2002 Korean (nationwide) health insurance database to obtain baseline data. Cohort participants were followed for 11 years, until 2013 [53]. However, this database does not include detailed information related to anesthetics, such as the specific medications or quantities of administered medication, which may influence postoperative cognitive outcomes or dementia. Moreover, other confounding factors such as drug consumption could not be controlled in our study. For example, our study did not include analyses of drug use affecting cognitive function, such as sedative-hypnotics, during the nine-year follow-up period. Thus, we lacked information on drugs that might to increase the risk of dementia or cognitive impairment, and this is a limitation of the work. In addition, the database did not include drug compliance and lifestyle factors, such as smoking and alcohol consumption, so these possible confounding factors could not be considered in our study. This was an inevitable limitation when using claim data without the information concerning actual drug administration. However, the principal limitation is the lack of data on possible confounding factors. It is thus possible that unmeasured confounders were in play. Thus, it may be difficult to conclude that our results reflect only an effect of GA. More precise result could be obtained when controlling for all possible confounding factors, indicating that a prospective cohort study that can control for all possible factors needs to be conducted.

Our large population study has a washout period for anesthesia of one year and excluded patients with additional anesthesia after the index period, to evaluate only the effect of GA during the index period. Furthermore, we perfectly matched the GA and non-GA groups using propensity scores for several variables, including age, sex, residence, household income, and comorbidities, and the effect of the matching variables on dementia showed similarity to previous studies. To prove our findings, we also performed sensitivity analyses. Their results revealed a higher frequency of GA event experience in patients with dementia. Future prospective studies should examine the association between the GA and dementia, including AD or VD in broader patient populations with long-term follow-up.

## 5. Conclusions

The present study investigated a possible link between GA and the development of dementia in middle-aged and older patients (≥55 years). We found that patients who underwent GA had a higher risk of developing dementia, with the risk being greater for female patients and those with combined comorbidities. Additionally, patients who had undergone GA showed higher risks for AD and VD, respectively. However, further studies are needed to elucidate the association/causality between GA and subsequent dementia.

## Figures and Tables

**Figure 1 jpm-11-01215-f001:**
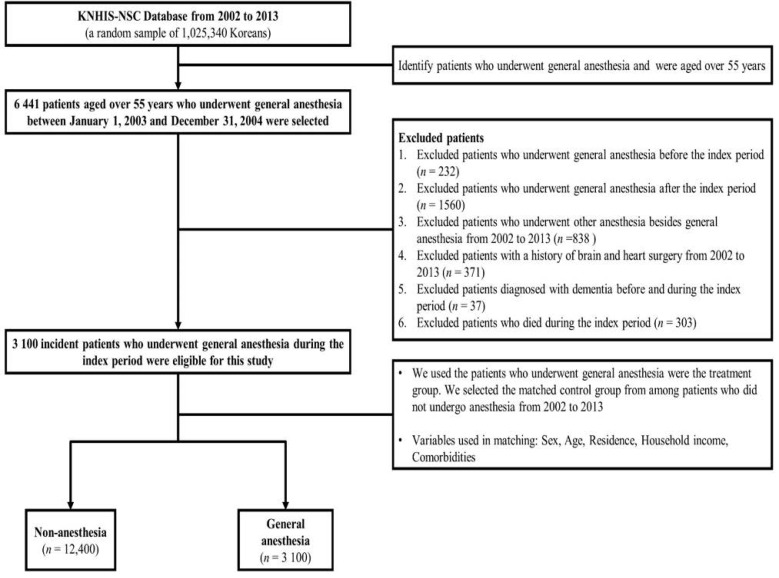
Schematic description of study’s design.

**Figure 2 jpm-11-01215-f002:**
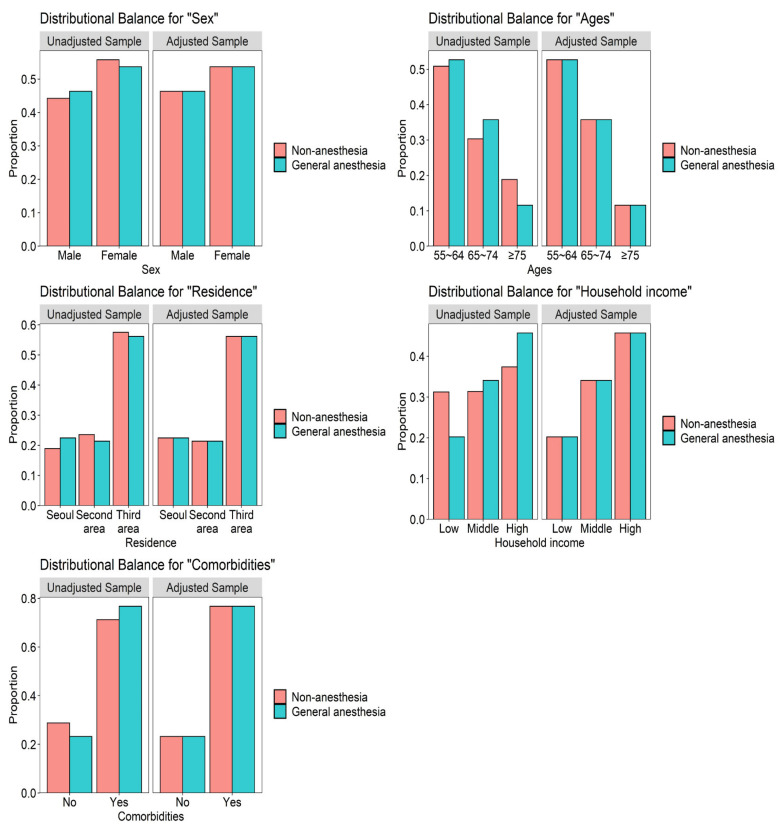
Balance plots of the five variables before and after matching.

**Figure 3 jpm-11-01215-f003:**
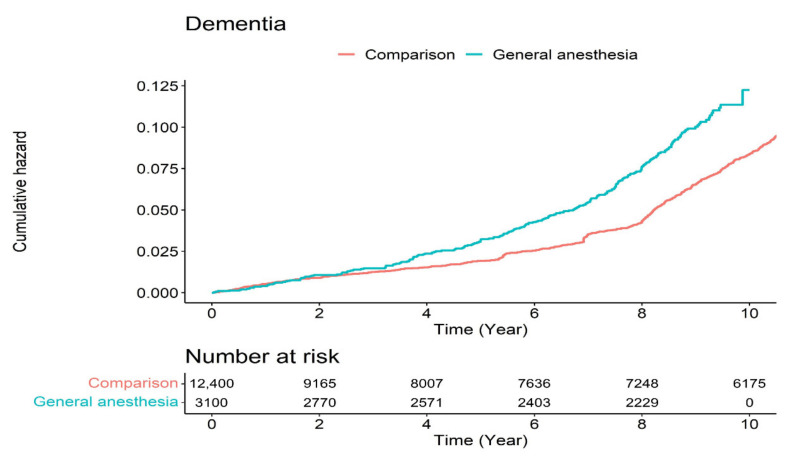
Cumulative hazard plot of dementia between the comparison group and those who underwent general anesthesia.

**Figure 4 jpm-11-01215-f004:**
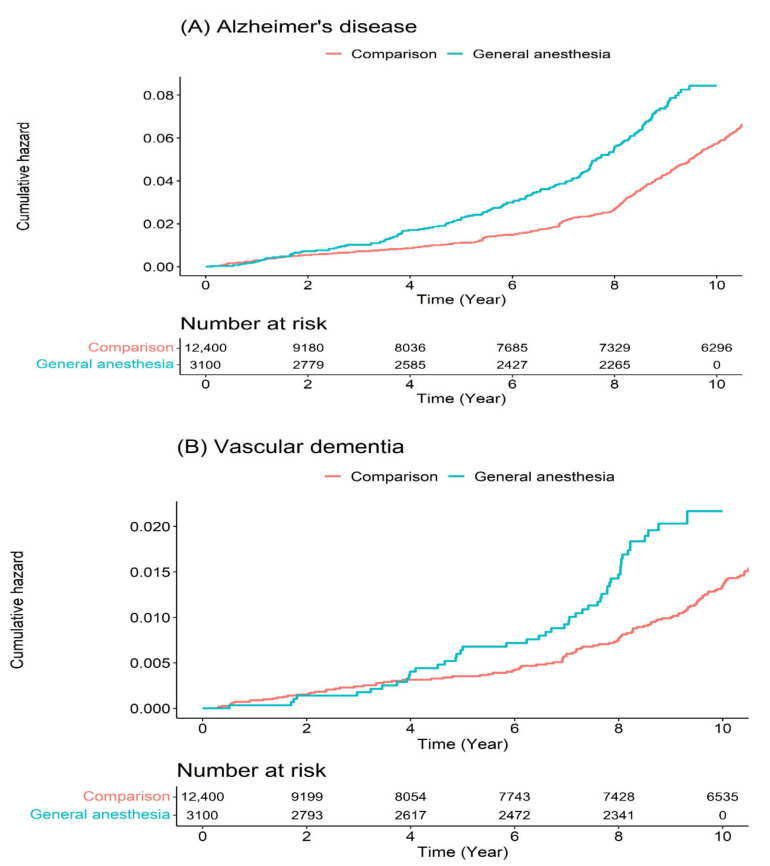
Cumulative hazard plots for specific diseases between patients who underwent general anesthesia and without anesthesia experience: (**A**) Alzheimer’s disease. (**B**) Vascular dementia.

**Table 1 jpm-11-01215-t001:** Characteristics of the study subjects.

Variables	Comparison (*n* = 12,400)	General Anesthesia (*n* = 3100)	*p* Value
Sex			1.000
Male	5744 (46.3%)	1436 (46.3%)	
Female	6656 (53.7%)	1664 (53.7%)	
Ages (years)			1.000
55–64	6532 (52.7%)	1633 (52.7%)	
65–74	4432 (35.7%)	1108 (35.7%)	
≥75	1436 (11.6%)	359 (11.6%)	
Residence			1.000
Seoul	2784 (22.5%)	696 (22.5%)	
Second area	2652 (21.4%)	663 (21.4%)	
Third area	6964 (56.2%)	1741 (56.2%)	
Household income			1.000
Low (0–30%)	2508 (20.2%)	627 (20.2%)	
Middle (30–70%)	4224 (34.1%)	1056 (34.1%)	
High (70–100%)	5668 (45.7%)	1417 (45.7%)	
Comorbidities			1.000
No	2880 (23.2%)	720 (23.2%)	
Yes	9520 (76.8%)	2380 (76.8%)	

Comparison, Subjects lacking anesthesia; Seoul, the largest metropolitan area; Second area, other metropolitan cities; Third area, other areas.

**Table 2 jpm-11-01215-t002:** Incidence per 1000 person-years and the HR (95% CI) of dementia.

Variables	N	Cases	Incidence	Unadjusted HR (95% CI)	Adjusted HR (95% CI)
Group					
Comparison	12,400	775	8.8	1.00 (ref.)	1.00 (ref.)
General anesthesia	3100	240	10.5	1.52 (1.30–1.76) ***	1.36 (1.16–1.58) ***
Sex					
Male	7180	364	7.9	1.00 (ref.)	1.00 (ref.)
Female	8320	651	9.9	1.20 (1.05–1.36) **	1.16 (1.02–1.32) *
Ages (years)					
55–64	8165	275	3.9	1.00 (ref.)	1.00 (ref.)
65–74	5540	554	15.4	4.25 (3.68–4.91) ***	3.95 (3.41–4.58) ***
≥75	1795	186	33.3	11.63 (9.63–14.05) ***	10.08 (8.29–12.26) ***
Residence					
Seoul	3480	196	7.5	1.00 (ref.)	1.00 (ref.)
Second area	3315	224	9.8	1.36 (1.12–1.65) **	1.27 (1.05–1.54) *
Third area	8705	595	9.6	1.31 (1.11–1.54) **	1.09 (0.93–1.28)
Household income					
Low (0–30%)	3135	354	13.2	1.00 (ref.)	1.00 (ref.)
Middle (30–70%)	5280	242	6.8	0.52 (0.44–0.61) ***	0.77 (0.65–0.91) **
High (70–100%)	7085	419	8.6	0.65 (0.56–0.75) ***	0.78 (0.68–0.91) **
Comorbidities					
No	3600	90	4.8	1.00 (ref.)	1.00 (ref.)
Yes	11,900	925	10.0	1.93 (1.55–2.40) ***	1.54 (1.24–1.92) ***

Seoul, The largest metropolitan area; Second area, other metropolitan cities; Third area, other areas. HR, hazard ratio; CI, confidence interval. * *p* < 0.05, ** *p* < 0.01, and *** *p* < 0.001.

**Table 3 jpm-11-01215-t003:** Hazard ratios of dementia by sex among the patients.

Sex	Male		Female	
	Comparison	General Anesthesia	Comparison	General Anesthesia
Dementia				
Unadjusted HR (95% CI)	1.00 (ref.)	1.33 (1.04–1.71) *	1.00 (ref.)	1.66 (1.37–2.00) ***
Adjusted HR (95% CI)	1.00 (ref.)	1.22 (0.95–1.58)	1.00 (ref.)	1.44 (1.19–1.75) ***

HR, hazard ratio; CI, confidence interval. * *p* < 0.05, and *** *p* < 0.001.

**Table 4 jpm-11-01215-t004:** Hazard ratios of dementia by the comorbidity status of the patients.

Comorbidities	No		Yes	
	Comparison	General Anesthesia	Comparison	General Anesthesia
Dementia				
Unadjusted HR (95% CI)	1.00 (ref.)	1.43 (0.90–2.28)	1.00 (ref.)	1.57 (1.34–1.84) ***
Adjusted HR (95% CI)	1.00 (ref.)	1.04 (0.64–1.68)	1.00 (ref.)	1.39 (1.18–1.64) ***

HR, hazard ratio; CI, confidence interval. *** *p* < 0.001.

**Table 5 jpm-11-01215-t005:** Incidence per 1000 person-years and the HRs (95% CI) for Alzheimer’s disease and vascular dementia.

Variables	N	Cases	Incidence	Unadjusted HR (95% CI)	Adjusted HR (95% CI)
Alzheimer’s disease					
Non-anesthesia	12,400	546	6.1	1.00 (ref.)	1.00 (ref.)
General anesthesia	3100	178	7.7	1.69 (1.41–2.02) ***	1.52 (1.27–1.82) ***
Vascular dementia					
Non-anesthesia	12,400	124	1.4	1.00 (ref.)	1.00 (ref.)
General anesthesia	3100	49	2.1	1.85 (1.31–2.61) ***	1.64 (1.15–2.33) **

HR, hazard ratio; CI, confidence interval. ** *p* < 0.01 and *** *p* < 0.001.

**Table 6 jpm-11-01215-t006:** Frequencies of general anesthesia between comparison and dementia groups.

	Comparison (*n* = 18,706)	Dementia (*n* = 9353)
General anesthesia		
0	16,098 (86.1%)	7844 (83.9%)
1	2147 (11.5%)	1204 (12.9%)
2	361 (1.9%)	239 (2.6%)
3	77 (0.4%)	50 (0.5%)
4	16 (0.1%)	9 (0.1%)
5	4 (0.0%)	4 (0.0%)
6	0 (0.0%)	2 (0.0%)
7	1 (0.0%)	0 (0.0%)
8	1 (0.0%)	0 (0.0%)
9	1 (0.0%)	0 (0.0%)
13	0 (0.0%)	1 (0.0%)

**Table 7 jpm-11-01215-t007:** Welch two sample *t*-test comparisons between the dementia and comparison groups.

	Frequency of General Anesthesia
Mean	
Comparison	0.171
Dementia	0.205
Standard deviation	
Comparison	0.479
Dementia	0.538
Levene test	
*F*	27.327
*df* (*v*1; *v*2)	1; 28,057
*p*-value	0.000
Welch Two Sample *t*-test	
*t*	−5.029
*df*	16,902
*p*-value	0.000
95% confidence interval	−0.046~−0.020

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
