# Peer review of "Longitudinal Study of the Association between General Anesthesia and Increased Risk of Developing Dementia"

_jpm, 2021, doi:10.3390/jpm11111215_

Round 1

Reviewer 1 Report

Thank you for permitting me to review this manuscript

The major concern 

As the authors suggested themselves there is probably baseline difference between patients having general anesthesia and those who hadn't since they had a pathology which necessited general anesthesia in addition regional anesthesia was not assessed at all some kind of comparison would have been interesting considering patients having the same pathology  anesthetized with a  regional anesthesia and assess  their outcomes in comparison to those having general anesthesia. This should at least be mentionned in the discussion section 

Minor concerns 

Figure 1 and  2 are difficult to read and need to be presented with a higher resolution 

Line 45 when the authors say that it is not possible to perform randomized study , they do not consider the possibility of regional anesthesia which cand be chosen in some situations , I think this  sentence should be deleted or another formula be chosen 

Exclusion criteria should be better explained 

Results: 

8.8  per 1000 vs 10.5 per 1000 the difference is 1.7 per 1000 I am not sure these results are clinically relevant giving that there is probably confounding factors 

these results should be cautiously discussed, I think the conclusion is a very weak difference and based in the fact that it is a retrospective analysis with propensity score matching drawing conclusion in a strong affirmative tone is over confidence 

Line 299-301 should be deleted since prospective study are necessary and caution is indicated 

Author Response

Oct 17, 2021

Reviewer 1

Journal of Personalized Medicine

Dear Reviewer 1,

Please find attached the revised version of our manuscript entitled “Longitudinal study of the association between general anesthesia and increased risk of developing dementia” (jpm-1395913).

We thank you for your thoughtful suggestions on your reading of the original version of our paper; most of the suggested changes have been incorporated into the revision.

All revisions are described in detail in the order mentioned in the review, following the reviewer’s critiques (in italics). We believe that the revisions have greatly improved the manuscript and hereby submit the revised version for consideration for publication.

Comments to author:

Thank you for permitting me to review this manuscript

We thank the reviewer for these comments and specific suggestions, which have improved our manuscript.

The major concern

As the authors suggested themselves there is probably baseline difference between patients having general anesthesia and those who hadn't since they had a pathology which necessited general anesthesia in addition regional anesthesia was not assessed at all some kind of comparison would have been interesting considering patients having the same pathology anesthetized with a regional anesthesia and assess their outcomes in comparison to those having general anesthesia. This should at least be mentionned in the discussion section

Thank you. We agree.

As recommended, we have added several sentences to the Discussion, as follows:

Also, a baseline difference probably exists between patients who underwent GA and not; the former exhibited pathologies that necessitated GA. We found it difficult to evaluate the effects of various surgeries, intra-operative events and perioperative complications such as hypoxia and hemodynamic instability. Although no effect of surgery per se on dementia risk was established, high-risk surgeries (such as cardiac surgery) have been reported to increase the risk of cognitive impairments, including delirium and postoperative cognitive dysfunction [51,52]. To overcome these limitations, it is necessary to evaluate the outcomes in patients with the same pathologies who underwent local anesthesia and GA.

(page 11, lines 296 – page 11, lines 304)

We have also added citations:

  1. Deiner S.; Silverstein J.H. Postoperative delirium and cognitive dysfunction. Br J Anaesth 2009, 103, Suppl 1. i41-i46, doi: 10.1093/bja/aep291.
  2. Cerejeira J.; Batista P.; Nogueira V.; Vaz-Serra A.; Mukaetova-Ladinska E.B. The stress response to surgery and postoperative delirium: evidence of hypothalamic-pituitary-adrenal axis hyperresponsiveness and decreased suppression of the gh/igf-1 axis. J Geriatr Psychiatry Neurol 2013.26.185-194. doi: 10.1177/0891988713495449.

Minor concerns

Figure 1 and 2 are difficult to read and need to be presented with a higher resolution

Thank you. We have revised Figures 1 and 2.

Line 45 when the authors say that it is not possible to perform randomized study, they do not consider the possibility of regional anesthesia which cand be chosen in some situations , I think this sentence should be deleted or another formula be chosen

Thank you. As recommended, we have deleted the sentence.

Exclusion criteria should be better explained

Thank you. We have modified the description of the exclusion criteria: “Additionally, patients were excluded if they (1) underwent GA before or after the index period, (2) were subjected to other types of anesthesia between 2002–2013, (3) had a history of brain or heart surgery between 2002–2013, (4) were diagnosed with dementia before the index period, or (5) died of any cause between 2002–2013.”

We have revised the Materials and Methods as follows:

Additionally, we excluded patients who: (1) underwent GA before or after the index period; (2) underwent other types of anesthesia from 2002 to 2013; (3) had a history of brain or heart surgery from 2002 to 2013; (4) were diagnosed with dementia before the index period; or (5) died of any cause from 2002 to 2013.

(page 2, lines 90 – page 2, lines 94)

Results:

8.8 per 1000 vs 10.5 per 1000 the difference is 1.7 per 1000 I am not sure these results are clinically relevant giving that there is probably confounding factors.

these results should be cautiously discussed, I think the conclusion is a very weak difference and based in the fact that it is a retrospective analysis with propensity score matching drawing conclusion in a strong affirmative tone is over confidence

Thank you. We agree. The Korea National Health Insurance Service–National Sample Cohort dataset focuses on medical claims and reimbursements; it is not a research dataset. It lacks information on procedures (thus both surgeries and any intraoperative or perioperative events). In addition, a baseline difference probably exists between those who underwent general anesthesia and not; the database does not capture this. Important limitations of our study include the lack of data on possible confounding factors and the possibility that unmeasured confounders were not considered. Thus, it may be difficult to conclude that our results reflect only the induction of general anesthesia.

We have revised the Discussion and Conclusions as follows:

However, the principal limitation is the lack of data on possible confounding factors; it is thus possible that unmeasured confounders were in play. Thus, it may be difficult to conclude that our results reflect only an effect of GA.

(page 11, lines 320 – page 11, lines 322)

The present study investigated a possible link between GA and the development of dementia in middle-aged and older patients (≥ 55 years). We found that patients who underwent GA had a higher risk of developing dementia, with the risk being greater for female patients and those with combined comorbidities. Additionally, patients who had undergone GA showed higher risks for AD and VD, respectively. However, further studies are needed to elucidate the association/causality between GA and subsequent dementia.

(page 11, lines 334 – page 12, lines 344)

Line 299-301 should be deleted since prospective study are necessary and caution is indicated

Thank you. We have deleted the sentence.

We have sought to address all issues raised by the reviewers. We are grateful for the constructive comments. We believe that our paper has been improved.

Sincerely,

Dong-Kyu Kim, MD, PhD & Young-Suk Kwon MD, PhD

Corresponding author: Dong-Kyu Kim, MD, PhD

Division of Big Data and Artificial Intelligence, Department of Otorhinolaryngology-Head and Neck Surgery, Chuncheon Sacred Heart Hospital, Hallym University College of Medicine (24253), 77, Sakju-ro, Chuncheon-si, Gangwon-do, Republic of Korea

Phone: 82-33-240-5180; Fax: 82-33-241-2909; E-mail: [email protected]

Corresponding author: Young-Suk Kwon M.D., Ph.D

Division of Big Data and Artificial Intelligence, Department of Anesthesiology and Pain Medicine, Chuncheon Sacred Heart Hospital, Hallym University College of Medicine (24253), 77, Sakju-ro, Chuncheon-si, Gangwon-do, Republic of Korea

Phone: 82-33-252-9970; Fax: 82-33-241-8063; E-mail: [email protected]

Reviewer 2 Report

The authors present interesting research for the field of neurology, however I have some suggestions or comments: 

The introduction is very sparse. An extension is needed where the study is contextualised mainly with literature reviews.
Methodology
Why were data collected up to 2013 and not more current data? The study is a little out of date. It is important to strengthen and justify this data both in the introduction and in the discussion.
results
Figures 1 and 2 are not legible, I have not been able to visualise them.
was drug consumption not controlled? This is an important limitation that may influence the results.
In the discussion, in limitations it is mentioned, but the answer is not justified in saying that "However, to over- come these issues, we enrolled only subjects who were 55 years of age or older and investigated the effect of GA in older age". I do not understand the age-drug concept, young patients with dementia also take drugs.

Author Response

Oct 17, 2021

Reviewer 2

Journal of Personalized Medicine

Dear Reviewer 2,

Please find attached the revised version of our manuscript entitled “Longitudinal study of the association between general anesthesia and increased risk of developing dementia” (jpm-1395913).

We thank you for your thoughtful suggestions; most of the suggested changes have been incorporated into the revision.

All revisions are described in detail in the order mentioned in the review, following the reviewer’s critiques (in italics). We believe that the revisions have greatly improved the manuscript and hereby submit the revised version for consideration for publication.

Comments to author:

The authors present interesting research for the field of neurology, however I have some suggestions or comments:

We thank the reviewer for the comments and specific suggestions, which have improved our manuscript.

The introduction is very sparse. An extension is needed where the study is contextualised mainly with literature reviews.

Thank you. We have added to and revised the Introduction as follows:

Dementia is the most common age-related disorder of the brain characterized by progressive cognitive and functional declines. AD is the most prevalent type of dementia; its predominance is expected to increase as in the aged population expands. Dementia imposes significant economic burdens on society. A better understanding of the relationship between dementia and surgery with anesthesia is critical.

(page 1, lines 39 – page 1, lines 43)

In vivo and in vitro studies suggest that anesthetics increase brain AD pathology, although such effects seem to be mediated by only certain anesthetics. Animal models support this perception by demonstrating associations between GA exposure and AD pathogenesis. Inhaled or intravenous anesthetics could promote the formation of Aβ plaques and neurofibrillary tangles [5–10]. Cell models have found that inhaled anesthetics induced apoptosis and increased β-amyloid protein levels. Commonly used inhaled anesthetics may promote AD neuropathogenesis [11,12]. Multiple anesthetic agents trigger hyperphosphorylation of the microtubule-associated protein tau, which forms neurofibrillary tangles in AD patients [7,13].

(page 1, lines 44 – page 2, lines 52)

Nationwide population-based cohort studies found significant associations between GA and dementia [17,35]. However, previous meta-analyses yielded conflicting results [27-29]. One meta-analysis reported a significant positive association between GA and AD [29], but others found only weak or no links [27, 28]. Thus, any association between anesthesia and dementia remains unclear; further investigation is needed.

(page 2, lines 57 – page 2, lines 61)

We have also added this citation:

  1. Zhen Y.; Dong Y.; Wu X.; Xu Z.; Lu Y.; Zhang Y.;Noron D.;Tian M.; Li S.; Xie Z. Nitrous Oxide Plus Isoflurane Induces Apoptosis and Increases β-Amyloid Protein Levels. Anesthesiology 2009; 111:741–52. doi: 10.1097/ALN.0b013e3181b27fd4.

Methodology

Why were data collected up to 2013 and not more current data? The study is a little out of date. It is important to strengthen and justify this data both in the introduction and in the discussion.

Thank you. The representative, nationwide population-based dataset contains information on medical service utilization by more than 1 million Koreans and was provided by the National Health Insurance Service of South Korea. The dataset was established in 2014; thus, this dataset was collected from 2002 to 2013, which included information on 1,025,340 nationally representative random subjects. That's why we could examine the association between general anesthesia and dementia until 2013 in this study.

We have revised the Introduction and Discussion as follows:

Therefore, we evaluated the risk of dementia a large population-based cohort that had undergone GA; we used a large dataset provided by the National Health Insurance Service (KNHIS) of South Korea.

(page 2, lines 64 – page 2, lines 66)

The KNHIS–NSC dataset, focuses on medical claims and reimbursements; this is not a research dataset. The representative, nationwide population-based dataset contains information on the medical service utilization by more than 1 million Koreans. The dataset was established in 2014; thus, this dataset was collected from 2002 to 2013, which comprised information from 1,025,340 nationally representative random subjects.

(page 11, lines 309 – page 11, lines 314)

results

Figures 1 and 2 are not legible, I have not been able to visualise them.

Thank you. We have revised Figures 1 and 2.

Was drug consumption not controlled? This is an important limitation that may influence the results.

We totally agreed with your opinion. So, we now describe this as a limitation, as follows:

However, this database does not include detailed information related to anesthetics such as the specific medications or quantities of administered medication, or drug use, which may influence postoperative cognitive outcomes or dementia. Thus, we lacked information on drugs that might to increase the risk of dementia or cognitive impairment; this is a limitation of the work.

(page 11, lines 314 – page 11, lines 318)

In the discussion, in limitations it is mentioned, but the answer is not justified in saying that "However, to over- come these issues, we enrolled only subjects who were 55 years of age or older and investigated the effect of GA in older age". I do not understand the age-drug concept, young patients with dementia also take drugs.

Thank you. We have deleted a portion of the Discussion and revised the remainder:

However, the principal limitation is the lack of data on possible confounding factors; it is thus possible that unmeasured confounders were in play. Thus, it may be difficult to conclude that our results reflect only an effect of GA.

(page 11, lines 318 – page 11, lines 322)

We have sought to address all issues raised by the reviewers. We are grateful for the constructive comments. We believe that our paper has been improved by the suggestions.

Sincerely,

Dong-Kyu Kim, MD, PhD & Young-Suk Kwon MD, PhD

Corresponding author: Dong-Kyu Kim, MD, PhD

Division of Big Data and Artificial Intelligence, Department of Otorhinolaryngology-Head and Neck Surgery, Chuncheon Sacred Heart Hospital, Hallym University College of Medicine (24253), 77, Sakju-ro, Chuncheon-si, Gangwon-do, Republic of Korea

Phone: 82-33-240-5180; Fax: 82-33-241-2909; E-mail: [email protected]

Corresponding author: Young-Suk Kwon M.D., Ph.D

Division of Big Data and Artificial Intelligence, Department of Anesthesiology and Pain Medicine, Chuncheon Sacred Heart Hospital, Hallym University College of Medicine (24253), 77, Sakju-ro, Chuncheon-si, Gangwon-do, Republic of Korea

Phone: 82-33-252-9970; Fax: 82-33-241-8063; E-mail: [email protected]

Round 2

Reviewer 1 Report

The authors have responed to my concerns and have imroved the manuscript

Author Response

Oct 24, 2021

Reviewer 1

Journal of Personalized Medicine

Dear Reviewer 1,

Please find attached the revised version of our manuscript entitled “Longitudinal study of the association between general anesthesia and increased risk of developing dementia” (jpm-1395913).

Comments to author:

The authors have responed to my concerns and have imroved the manuscript

We thank the reviewer’s comments.

We have sought to address all issues raised by the reviewers. We are grateful for the constructive comments. We believe that our paper has been improved.

Sincerely,

Dong-Kyu Kim, MD, PhD & Young-Suk Kwon MD, PhD

Corresponding author: Dong-Kyu Kim, MD, PhD

Division of Big Data and Artificial Intelligence, Department of Otorhinolaryngology-Head and Neck Surgery, Chuncheon Sacred Heart Hospital, Hallym University College of Medicine (24253), 77, Sakju-ro, Chuncheon-si, Gangwon-do, Republic of Korea

Phone: 82-33-240-5180; Fax: 82-33-241-2909; E-mail: [email protected]

Corresponding author: Young-Suk Kwon M.D., Ph.D

Division of Big Data and Artificial Intelligence, Department of Anesthesiology and Pain Medicine, Chuncheon Sacred Heart Hospital, Hallym University College of Medicine (24253), 77, Sakju-ro, Chuncheon-si, Gangwon-do, Republic of Korea

Phone: 82-33-252-9970; Fax: 82-33-241-8063; E-mail: [email protected]

Reviewer 2 Report

Although the authors have answered all the questions raised in the previous review, I still do not understand why the data collection has not been updated. The data provided may be out of date now in 2021.
On the other hand, there is an important limitation in this research that has not been controlled for, and that is the control of drug consumption.

Author Response

Oct 24, 2021

Reviewer 2

Journal of Personalized Medicine

Dear Reviewer 2,

Please find attached the revised version of our manuscript entitled “Longitudinal study of the association between general anesthesia and increased risk of developing dementia” (jpm-1395913).

We thank you for your thoughtful suggestions; most of the suggested changes have been incorporated into the revision.

All revisions are described in detail in the order mentioned in the review, following the reviewer’s critiques (in italics). We believe that the revisions have greatly improved the manuscript and hereby submit the revised version for consideration for publication.

Comments to author:

Although the authors have answered all the questions raised in the previous review, I still do not understand why the data collection has not been updated. The data provided may be out of date now in 2021.

We thank the reviewer for the comments, which have improved our manuscript.

The National Health Insurance Service–National Sample Cohort (NHIS-NSC) is a population-based cohort established by the National Health Insurance Service in South Korea. The sole purpose of constructing the cohort was to provide public health researchers and policy makers with representative, useful information regarding citizens’ utilization of health insurance and health examinations. A total of 1,025,340 participants of the cohort, 2.2% of the total eligible population, were randomly sampled from the 2002 Korean (nationwide) health insurance database to obtain baseline data. Cohort participants were followed for 11 years, until 2013. During the follow-up period, a representative sample of newborns (age 0) was added annually and deceased or emigrated participants were excluded. In 2013, the database included 1,014,730 participants.

This dataset was built in 2014 and distributed to researchers. Dataset for subsequent cohorts has not yet been released.

We have added to and revised the Discussion as follows:

A total of 1,025,340 participants of the cohort, 2.2% of the total eligible population, were randomly sampled from the 2002 Korean (nationwide) health insurance database to obtain baseline data. Cohort participants were followed for 11 years, until 2013 [53].

(page 11, lines 302 – page 11, lines 306)

We have also added citations:

  1. J Lee.; JS Lee.; SH Park.; SA Shin.; KW Kim. Cohort Profile: The National Health Insurance Service–National Sample Cohort (NHIS-NSC), South Korea. Int J Epi 2017, e15(1–8), doi: 10.1093/ije/dyv319

On the other hand, there is an important limitation in this research that has not been controlled for, and that is the control of drug consumption.

We totally agreed with reviewer’s comments. The NHIS database we used did not include detailed information related to anesthetics such as the specific medications or quantities of administered medication which may influence postoperative cognitive outcomes or dementia. In addition, other confounding factors such as drug consumption could not be controlled in our study. For example, our study did not include analyzes of drug use affecting cognitive function, such as sedative-hypnotics, during the 9-year follow-up period. Also, the database did not include drug compliance and lifestyle factors, such as smoking and alcohol consumption, so these possible confounding factors could not be considered in our study. This was an inevitable limitation when using claim data without the information about actual drug administration. Thus, a more precise result could be obtained when controlling for all possible confounding factors, indicating that a prospective cohort study that can control for all possible factors needs to be conducted.

So, we have added to and revised the Discussion as follows:

However, this database does not include detailed information related to anesthetics such as the specific medications or quantities of administered medication, which may influence postoperative cognitive outcomes or dementia. Also, other confounding factors such as drug consumption could not be controlled in our study. For example, our study did not include analyzes of drug use affecting cognitive function, such as sedative-hypnotics, during the 9-year follow-up period.

(page 11, lines 307 – page 11, lines 312)

Also, the database did not include drug compliance and lifestyle factors, such as smoking and alcohol consumption, so these possible confounding factors could not be considered in our study. This was an inevitable limitation when using claim data without the information about actual drug administration.

(page 11, lines 314 – page 11, lines 317)

More precise result could be obtained when controlling for all possible confounding factors, indicating that a prospective cohort study that can control for all possible factors needs to be conducted.

(page 11, lines 320 – page 11, lines 322)

We have sought to address all issues raised by the reviewers. We are grateful for the constructive comments. We believe that our paper has been improved by the suggestions.

Sincerely,

Dong-Kyu Kim, MD, PhD & Young-Suk Kwon MD, PhD

Corresponding author: Dong-Kyu Kim, MD, PhD

Division of Big Data and Artificial Intelligence, Department of Otorhinolaryngology-Head and Neck Surgery, Chuncheon Sacred Heart Hospital, Hallym University College of Medicine (24253), 77, Sakju-ro, Chuncheon-si, Gangwon-do, Republic of Korea

Phone: 82-33-240-5180; Fax: 82-33-241-2909; E-mail: [email protected]

Corresponding author: Young-Suk Kwon M.D., Ph.D

Division of Big Data and Artificial Intelligence, Department of Anesthesiology and Pain Medicine, Chuncheon Sacred Heart Hospital, Hallym University College of Medicine (24253), 77, Sakju-ro, Chuncheon-si, Gangwon-do, Republic of Korea

Phone: 82-33-252-9970; Fax: 82-33-241-8063; E-mail: [email protected]
